(c) Author(s) 2017. CC BY 4.0 License.

# Global teleconnectivity structures of the El Niño-Southern **Oscillation and large volcanic eruptions – An evolving network** perspective

Tim Kittel<sup>1,2</sup>, Catrin Ciemer<sup>1,2</sup>, Nastaran Lotfi<sup>3</sup>, Thomas Peron<sup>3</sup>, Francisco Rodrigues<sup>3</sup>, Jürgen Kurths<sup>1,2</sup>, and Reik V. Donner<sup>1</sup>

<sup>1</sup>Potsdam Institute for Climate Impact Research, Telegrafenberg A31, 14473 Potsdam, Germany

<sup>2</sup>Department of Physics, Humboldt University Berlin, Newtonstraße 15, 12489 Berlin, Germany

<sup>3</sup>Institute of Mathematics and Computer Science, University of São Paulo, Avenida Trabalhador Sao-carlense, 400-Centro, 13566-590 Sao Carlos, Brazil

Abstract. Recent work has provided ample evidence that global climate dynamics at time-scales between multiple weeks and several years can be severely affected by the episodic occurrence of both, internal (climatic) and external (non-climatic) perturbations. Here, we aim to improve our understanding on how regional to local disruptions of the "normal" state of the global surface air temperature field affect the corresponding global teleconnectivity structure. Specifically, we present an approach to

- quantify teleconnectivity based on different characteristics of functional climate network analysis. Subsequently, we apply this 5 framework to study the impacts of different phases of the El Niño-Southern Oscillation (ENSO) as well as the three largest volcanic eruptions since the mid 20th century on the dominating spatio-temporal co-variability patterns of daily surface air temperatures. Our results confirm the existence of global effects of ENSO which result in episodic breakdowns of the hierarchical organization of the global temperature field. This is associated with the emergence of strong teleconnections. At more
- regional scales, similar effects are found after major volcanic eruptions. Taken together, the resulting time-dependent patterns 10 of network connectivity allow a tracing of the spatial extents of the dominating effects of both types of climate disruptions. We discuss possible links between these observations and general aspects of atmospheric circulation.

# 1 Introduction

The empirical analysis of climate data is fundamental to understand the evolution and develop more accurate methods for 15 forecasting of climate phenomena like El Niño. Typically, such data sets comprise time series representing temperature, precipitation or other climate variables observed at distinct locations distributed around the globe. Their common properties include long-range spatial and often also temporal correlations (Fraedrich and Blender, 2003), interactions at and among multiple scales (Paluš, 2014) and nonlinearity (Dijkstra, 2013). With the Earth's surface being subdivided into regions for which individual "grid points" and associated localized records of climate variability are considered representative, the evolution of

the climate system can be approximately described by a high-dimensional multivariate time series composed of a multitude of 20 interdependent signals.

While the analysis of such big climate data sets has been traditionally attempted by means of classical statistical approaches like empirical orthogonal function or maximum covariance analysis (von Storch and Zwiers, 2003), it has recently been realized that these methods exhibit fundamental intrinsic limitations, including their linearity and associated condition of pairwise orthogonal patterns (Gámez et al., 2004). As a consequence, the traditional view that the corresponding decompositions of global spatio-temporal co-variability patterns actually provide dynamical (or at least statistical) modes that unambiguously coincide with specific key climatic processes has been abandoned (Monahan et al., 2009). Taken together, there is growing evidence that the application of traditional linear methods of signal processing and the climatic interpretation of their results are severely affected by the dynamical complexity of the involved processes.

During the last years, complex network representations of climate variability have been developed (Tsonis et al., 2006;

- Donges et al., 2009a, b; Tsonis et al., 2011; Tsonis and Swanson, 2012; Steinhaeuser et al., 2012; Peron et al., 2014; Ciemer et al., 2017) and demonstrated to provide a suitable approach for relieving some of the aforementioned concerns (Donges et al., 2015b). In this nonlinear statistical framework, referred to as *functional climate network analysis*, the individual grid points or cells are considered as nodes of a spatially embedded graph. Connections among these nodes are established according to similarities between the individual (local) climate time series (Tsonis et al., 2006; Donges et al., 2009b; Donner et al., 2017).
- By construction, the network structures thus obtained highlight essential statistical interrelationships among spatio-temporal climate data (Donges et al., 2009b).

The application of functional climate networks has already provided several important insights. For instance, centrality measures, such as betweenness centrality, have been found to serve as tracers of global circulation patterns in the atmosphere and oceans (Donges et al., 2009a). Moreover, climate networks have been used to identify dipole patterns which represent pressure

- anomalies of opposite polarity appearing in two different regions simultaneously (Kawale et al., 2011). The study of the coupling structure between interdependent climate variables (Donges et al., 2011), the temporal evolution and teleconnections of the North Atlantic Oscillation (Guez et al., 2012, 2013), the distinction of different types of El Niño phases (Radebach et al., 2013; Wiedermann et al., 2016) and the prediction of the latter (Ludescher et al., 2013, 2014) have also been subjects of corresponding recent investigations. Many of the aforementioned methodological achievements have been integrated in open source
- software packages (Donges et al., 2015a) contributing to the increasing use of functional network analysis in climatological studies (Donner et al., 2017).

One rather fundamental property of large networks is their (possibly hierarchical) organization in terms of communities – an aspect that has also been addressed recently in the context of key patterns in climate data (Tsonis et al., 2011). Here, a community is a subset of densely connected nodes which exhibit only few interactions with the rest of the network (Newman,

2006b; Fortunato, 2010). In a climate network context, communities would ideally have some climatological interpretation. Specifically, Tsonis et al. (2011) argued that each community in a climate network should be considered as a subsystem which operates relatively independent of the other communities. Besides corresponding connectivity structures in individual climate variables, community detection algorithms (Fortunato, 2010) can also be used to detect multi-variable clusters (Steinhaeuser et al., 2010).

**Figure 1.** Main regions of interest used within this paper. Sets of blue dots labelled with "El Chichon", "Agung" and "Pinatubo" indicate grid points within a  $5^{\circ}$  radius around the corresponding volcanoes. The numbers 3, 3.4 and 4 identify the corresponding Nino regions (cf. Tab. 1) commonly used for defining characteristic indices of ENSO variability. The region "ENSO-big" will be removed from the complete global data set when analyzing the spatial imprints of volcanic eruptions to ensure that ENSO-related effects are excluded.

In this paper, we analyze global surface air temperature data in terms of functional climate networks and demonstrate the close relationship between El Niño and La Niño episodes as well as strong volcanic eruptions on the one hand, and temporal changes in the modular organization of the resulting networks on the other hand. For this purpose, we study the teleconnectivity structure in the climate system in terms of spatial fields of two network properties that represent the number of strong statistical connections, as well as the average spatial distance between the connected grid points. In addition, the associated temporal

variations are traced by some scalar-valued global network characteristics.

The remainder of this paper is organized as follows: Sect. 2 provides brief information on the climatological background of ENSO and volcanic eruptions as the two types of major climatic disruptions studied in this work. The data and methodology used in this work are described in detail in Sect. 3. Finally, our results are presented and discussed in Sect. 4, followed by concluding remarks.

5

10

# 2 Climatological background

#### 2.1 El Niño–Southern Oscillation

Among the dominant teleconnectivity patterns in the global climate system, the El Niño–Southern Oscillation (ENSO) is the probably most remarkable phenomenon in terms of both, the magnitude of associated variations in sea-surface temperature

© Author(s) 2017. CC BY 4.0 License.

| region   | latitudes                       | longitudes                       |
|----------|---------------------------------|----------------------------------|
| Nino1+2  | $10^{\circ}$ S - $0^{\circ}$ N  | $90^{\circ}W - 80^{\circ}W$      |
| Nino3    | $5^{\circ}S - 5^{\circ}N$       | $150^{\circ}$ W - $90^{\circ}$ W |
| Nino4    | 5°S - 5°N                       | $160^{\circ}E - 150^{\circ}W$    |
| Nino3.4  | 5°S - 5°N                       | $170^{\circ}W - 120^{\circ}W$    |
| ENSO-big | $30^{\circ}$ S - $10^{\circ}$ N | $180^{\circ}W - 60^{\circ}W$     |

Table 1. Overview on different regions commonly used for defining characteristic temperature-based indices associated with ENSO variability. In addition, we include the definition of the "ENSO-big" region studied in this work, which corresponds to the region that is discarded in our analyses of the impacts of strong volcanic eruptions on global temperature teleconnectivity.

(SST) and sea-level pressure, as well as its specific impacts on different aspects of regional climate variability worldwide (Trenberth, 1997). During the positive phase (El Niño) of this complex oscillation of the coupled atmosphere–ocean system in the tropical Pacific, the eastern tropical Pacific exhibits some anomalous warming with respect to "normal" mean conditions, while the negative phase (La Niña) is characterized by a corresponding cooling. In comparison with the normal mean

- climatology, this surface temperature anomaly results in marked shifts of key atmospheric pressure systems, modifying the large-scale circulation and, thus, leading to prominent shifts of, e.g., precipitation patterns. It has been shown that effects of both ENSO phases can be observed in remote regions including North and South America, Africa, the Indian subcontinent, and even Antarctica (Ropelewski and Halpert, 1987; Dai and Wigley, 2000; Neelin et al., 2003; Turner, 2004; Clarke, 2008; Sarachik and Cane, 2010).
- The long-term variability of ENSO is characterized by some irregular oscillations with a period of 2 to 7 years and remarkable variations in the associated characteristic frequencies and amplitudes of both, temperature and pressure anomalies. Following its prominent spatial structures in tropical SST and sea level pressure, ENSO is commonly traced by indices that take up the variability of the aforementioned observables in some key region of the tropical Pacific ocean. Notably, a set of indices has been defined in terms of average SST anomalies taken over distinct regions in the eastern and central tropical Pacific, referred
- to as Nino1+2, Nino3, Nino4 and Nino3.4, respectively (Trenberth and Stepaniak, 2001) (see Fig. 1 and Tab. 1). In this work, we will utilize the so-called Ocean Niño Index (ONI) for differentiating between different phases of ENSO. It is defined as the running three-month mean SST anomaly for the Niño 3.4 region (5°N–5°S, 120°–170°W) in comparison with centered 30-year base periods that are updated every 5 years (NCEP, 2017). When the ONI exceeds 0.5°C for at least five consecutive months, the corresponding situation is classified as an El Niño, and the magnitude of the ONI is taken as an indicator of the
- strength of the corresponding event. In turn, if the ONI drops below  $-0.5^{\circ}$ C for at least five consecutive months, this indicates a La Niña episode.

In the last years, it has been recognized that the commonly observed spatial patterns associated with El Niño (as well as La Niña) related SST anomalies are far from being homogeneous across the set of observed events. Consequently, it has been suggested to further distinguish both phases into two respective flavours (see Wiedermann et al., 2016, and references therein).