# Peer review of "Global teleconnectivity structures of the El Niño-Southern Oscillation and large volcanic eruptions – An evolving network perspective"

_Nonlinear Processes in Geophysics, 2017_

## Referee Comment (RC1) · Anonymous Referee #1 · 28 Dec 2017

Using functional network theory applied to the global NCEP reanalysis SAT data set, the authors aim to quantify teleconnectivity changes associated with ENSO variability (El Nino and La Nina) and due to several 20th century volcanic eruptions. The new aspects are the combined use of the transitivity, modularity and global averaged link distance measures as well as regionalized versions of the degree and average distance measures.

The use of functional network analysis is relatively new in climate. However, many papers have already appeared on more descriptive aspects of the reconstructed networks. One would expect that new papers would also attempt to connect such results

better to phenomena deduced from more classical analysis of observations and climate models. The present paper could particularly serve such a connection but, as it is written, it does not extend much above a description of the results on the network level.

My major point is therefore that the authors should make a better effort to connect this (network) description to physical phenomena found in observations and models. This requires a substantial literature search and a rewrite of the sections 2, 4 and 5. In addition, I have some other remarks that the authors may use to improve the paper.

Remarks:

1. p2, l6-l8: That statement makes no sense (to me). Please rephrase and add a reference.

2. p2, l32-33: References missing here are the papers by Fountalis et al., Clim. Dynamics, (2015) and Tantet and Dijkstra, Earth System Dynamics, (2014) and these should be put in the right context (i.e. application of community detection algorithms).

3. p4, l10: This is not the long-term variability of ENSO (which is decades and longer) but the dominant interannual time scale.

4. Section 2: This section should focus more on known responses (teleconnections) to extreme phases of ENSO variability and responses to volcanic eruptions.

5. Section 3: This section can be shortened substantially by moving most of section 3.2 to an appendix (it has presented many times before). Section 3 could then focus on the data, how they are filtered and what the network quantities computed (with reference to the appendix) mean.

6. p8, l9: On which criteria is the choice 0.5% based and how sensitive is a result such as in Fig. 2 dependent on this choice?

7. section 3.3: Maybe good to mention here that the subset is taken after the network

is reconstructed (and no new network is reconstructed only based on the subset of nodes).

8. p13, l15-19: Just when it gets interesting ... No further analysis is done here, but this should be much better clarified in a revised version of the paper.

9. p13, l20-25: There is also a second (weak) minimum in the modularity so it should be much better explained what the different behavior of Q and T indicates here (also in lines 30-35) in terms of teleconnections (e.g. more midlatitude like such as the PNA pattern or more global along tropical latitudes). This should also be consistent with the patterns found in Fig. 3.

10. section 4.1.3: One would expect that the Nino3 and Nino3.4 regions are each highly connected so what new information is extracted from the results in Fig. 4 regarding teleconnections?

11. section 4.2: These results are interesting, but for their interpretation it would help to look at which specific changes have been observed (changes in atmospheric circulation, convection, etc.) during these periods.

12. p21, l25-27: The statement does not make sense (to me). Please rephrase.

---

## Referee Comment (RC2) · Anonymous Referee #2 · 28 Dec 2017

The authors investigate the connectivity properties of the temperature network with emphasis on the impacts of ENSO events and volcanic eruptions on the structure of the network.

While I have no problem with the methods and the analysis, I am not impressed with the results. That El Nino and La Nina have effects and long-range correlations on the temperature field is not news. The same goes with volcanic eruptions. I, therefore, agree with anonymous reviewer #1 that more has to be done for this paper to be an original paper.
* * *
2017-69, 2017.

---

## Author Comment (AC1) · 3 Apr 2018

We are extremely grateful to both reviewers for their thorough reading of our manuscript and recommendations provided on how to improve the presented material. We are now ready to prepare a revised version of our paper along the lines of our detailed response to the raised criticisms as given below, and would appreciate if we were given the chance to resubmit an accordingly modified paper that addresses the specific requests of both reviewers to the largest possible extent.

**Anonymous Referee #1**

*Using functional network theory applied to the global NCEP reanalysis SAT data set, the authors aim to quantify teleconnectivity changes associated with ENSO variability (El Nino and La Nina) and due to several 20th century volcanic eruptions. The new aspects are the combined use of the transitivity, modularity and global averaged link distance measures as well as regionalized versions of the degree and average distance measures.*

*The use of functional network analysis is relatively new in climate. However, many papers have already appeared on more descriptive aspects of the reconstructed networks. One would expect that new papers would also attempt to connect such results better to phenomena deduced from more classical analysis of observations and climate models. The present paper could particularly serve such a connection but, as it is written, it does not extend much above a description of the results on the network level.*

We agree with the reviewer that functional climate network analysis, despite originally suggested by Tsonis and Roebber in their seminal paper in 2004 and later taken up by various research groups, is still a rather young method that requires both better theoretical understanding of the methodology and proof of added value in climatological applications. While the reviewer appears to put a focus on the second aspect, the original goal of our study was rather to put forward a novel combination of network properties that has not been employed in previous studies. In this regard, we primarily aim here at putting forward the methodological aspect by addressing the relevant question which kind of climate phenomena and processes are best captured by which type of network properties. In our opinion, this needs to be understood first (in a more systematic way than in previous studies) before a possible detailed evaluation and interpretation of the obtained results. In fact, this is also why we have chosen "Nonlinear Processes in Geophysics" as a forum for publishing our corresponding results: the manuscript is mainly targeted to the community working on new methodological approaches rather than "classical" climatology.

Accordingly, our manuscript shall demonstrate that a nonlinear approach using functional climate networks can sensibly distinguish different types of large-scale climate disruptions. In this context, we have focused on the structural distinction between impact patterns due

to ENSO events and volcanic eruptions based upon different network characteristics. An inter-comparison with results from classical analysis techniques (as discussed, e.g., in Donges et al. (2015) or Wiedermann et al. (2017)) has not been intended and is in our opinion also outside the scope of our manuscript, even though we generally think that this is desirable (as a second step) and might provide relevant information to the field of climatology in a more general context. In fact, the latter is a subject of our ongoing work as well as research projects pursued by a number of groups worldwide.

In the above spirit, we understand our analysis of large-scale climate impacts of ENSO phases and volcanic eruptions as a primarily methodological analysis, since some detailed analysis on specific (atmospheric) processes based upon observational and, to an even greater extent, model data using classical statistical analysis techniques has been the focus of a variety of previous studies. We explicitly do not aim here to validate already known, or identify or suggest new mechanisms associated with the reorganization of the spatio-temporal correlation structure of global surface air temperatures, but rather to describe the subtle effects that are manifested in the associated climate network properties. In particular, our current work takes up some open research questions from the previous publications by Radebach et al. (2013) and Wiedermann et al. (2016) that have already shown that some global climate network properties are sensitive to both, large-scale coherent warming/cooling patterns associated with (strong, commonly East Pacific) El Nino and La Nina phases as well as strong volcanic eruptions. We argue that our manuscript should be seen in the corresponding context as partly closing the existing knowledge gap at the methodological side rather than providing fundamentally new information on the climatological aspects. We agree, however, that a more systematic exploitation of our results in the context of specific processes, possibly combining observational and model data, would be desirable and should be emphasized more clearly as an important subject of future studies.

*My major point is therefore that the authors should make a better effort to connect this (network) description to physical phenomena found in observations and models. This requires a substantial literature search and a rewrite of the sections 2, 4 and 5.*

As we have detailed above, we agree that this comment is highly valuable. However, providing an excessive literature review on associated processes would bear the risk of turning our manuscript into an extensively long paper that would lose its methodological focus and, by this means, might become less directly accessible to the readership of Nonlinear Processes in Geophysics. We agree that clarifying our research agenda and also highlighting links to known physical phenomena from observational and model data is desirable, and are ready to revise the mentioned sections accordingly.

*In addition, I have some other remarks that the authors may use to improve the paper.*

**Remarks:**

*1. p2, l6-l8: That statement makes no sense (to me). Please rephrase and add a reference.*

Here, the reviewer appears to refer to our statement that "there is growing evidence that the application of traditional linear methods of signal processing and the climatic interpretation of their results are severely affected by the dynamical complexity of the involved processes". We agree that this statement might appear somewhat bold in its generality. Regarding classical EOF analysis as the most direct linear analog to functional climate network analysis, Monahan et al. (2009), cited in the previous sentence, has provided arguments that the classical approach can be misleading in terms of both, its methodological limitations (linearity assumption) as well as interpretation (as providing relevant dynamical and/or statistical modes of climate variability). From the large field of time series analysis, similar points could be made (i.e., classical analysis methods like power spectra miss relevant information due to the nonlinear nature of climate variability). We will attempt to clarify our statement accordingly.

*2. p2, l32-33: References missing here are the papers by Fountalis et al., Clim. Dynamics, (2015) and Tantet and Dijkstra, Earth System Dynamics, (2014) and these should be put in the right context (i.e. application of community detection algorithms).*

We thank the reviewer for pointing our attention to these important studies and will include them in the revised manuscript.

*3. p4, l10: This is not the long-term variability of ENSO (which is decades and longer) but the dominant interannual time scale.*

We agree that the corresponding sentence has been phrased in some potentially misleading way, and will reformulate our statement accordingly.

*4. Section 2: This section should focus more on known responses (teleconnections) to extreme phases of ENSO variability and responses to volcanic eruptions.*

As detailed in our response to the general comments above, the focus of our work is rather on better understanding the employed analysis method than on highlighting specific climate processes. We agree that it would be worthwhile discussing the corresponding phenomena in greater detail, but providing an extensive amount of specific information on the climatic processes would make our manuscript lose its designated focus. As a trade-off, we suggest adding some further discussion and references to Section 2, while not providing an exhaustive treatment of the two topics of (extreme) ENSO teleconnectivity and large-scale climate impacts of volcanic eruptions.

*5. Section 3: This section can be shortened substantially by moving most of section 3.2 to an appendix (it has presented many times before). Section 3 could then focus on the data, how they are filtered and what the network quantities computed (with reference to the appendix) mean.*

Along with our general considerations as presented above, we respectfully disagree with this suggestion. Making our manuscript a primarily methodological reference and discussing the specific type of information provided by different climate network characteristics requires a clear introduction and thorough description of the analysis methods and specific network properties. This is particularly important since other studies utilizing climate networks have made distinctively different choices of key methodological options (e.g., keeping a fixed correlation threshold instead of a fixed network connectivity in the works by Yamasaki et al. (2008), Gozolchiani et al. (2009) or Ludescher et al. (2013, 2014)), which can lead to apparently contradictory results if only looking at the behavior of network characteristics from a purely descriptive viewpoint (see Radebach et al. (2013) for a corresponding brief discussion). In this context, to maintain the readability of our present work, we do not see much potential to reduce the overall volume of the presented material. Moving vast parts of Section 3.2 to an additional appendix would only shift the problem, but might make the manuscript as a whole even harder to read.

*6. p8, l9: On which criteria is the choice 0.5% based and how sensitive is a result such as in Fig. 2 dependent on this choice?*

The selection of a link density of 0.5% is based on detailed investigations in previous studies by Radebach et al. (2013) and Wiedermann et al. (2016). In general, it is a heuristic choice that has been found to provide a reasonable trade-off in the context of making the network characteristics relatively sensitive to the emergence of "localized structures" in the climate network. This does not mean that other choices are generally worse, but for the purpose of this study (tracing subtle changes in the global spatio-temporal correlation pattern of surface air temperatures), too high link densities might blur the effects of interest while too

low densities could make the network decompose into disconnected components, thereby prohibiting the application of the selected network characteristics.

In order to illustrate the choice of 0.5%, as well as the qualitative robustness of some of our main findings, we have repeated our analysis with two different link densities of 0.1% and 2.5%, respectively, and did not find strong differences to our previous analysis (see Fig. S1 and Fig. S2 below for some results). We will add some brief discussion of this fact in our revised manuscript and are ready to provide further details in some Supplementary Material if requested.

[Figure]

Fig. S1: Network transitivity (a) and modularity (b) evolution for a link density of 0.1%.

[Figure]

Fig. S2: As Fig. S1 for a link density of 2.5%.

*7. section 3.3: Maybe good to mention here that the subset is taken after the network is reconstructed (and no new network is reconstructed only based on the subset of nodes).*

We thank the reviewer for making us aware of this possible misunderstanding and will make it clearer in the revised manuscript.

*8. p13, l15-19: Just when it gets interesting ... No further analysis is done here, but this should be much better clarified in a revised version of the paper.*

We fully agree that a lot more could be done here. We will clarify the scope of the present work, as well as further directions of research originating from the results presented in the mentioned part of our manuscript in a thoroughly revised version of our manuscript.

*9. p13, l20-25: There is also a second (weak) minimum in the modularity so it should be much better explained what the different behavior of Q and T indicates here (also in lines 30-35) in terms of teleconnections (e.g. more midlatitude like such as the PNA pattern or more global along tropical latitudes). This should also be consistent with the patterns found in Fig. 3.*

We are grateful for this comment and will clarify the specific meaning of characteristic behaviors of Q and T, respectively. Regarding specific teleconnections, the global network properties transitivity and modularity themselves do not allow attributing corresponding effects to specific regional patterns, which would rather call for the associated consideration of local (grid point-wise) characteristics. Specifically, the least that would be required for a corresponding attribution would be "regionalized" versions of the global maps from Fig. 3. However, this will require the educated choice of some "source" region of interest. We are currently performing corresponding detailed analyses, but hesitate to present them in detail in some revised version of the paper since this would imply adding a lot of additional material to an already very long manuscript. In any case, a corresponding qualitative discussion will be included in a possible revision.

*10. section 4.1.3: One would expect that the Nino3 and Nino3.4 regions are each highly connected so what new information is extracted from the results in Fig. 4 regarding teleconnections?*

In the last years, there has been an intense discussion on different types of ENSO episodes with the core centers of action located either in the Eastern or more central tropical Pacific (as also discussed by Radebach et al. (2013) and Wiedermann et al. (2016) in the context of climate network signatures). The Nino3 and Nino4 regions provide convenient choices to highlight these different types, in our case, regarding their respective teleconnectivity structure. Besides a general similarity between the results obtained for both focus regions, this may temporarily lead to prominent differences as, e.g., visible during the year 2015.

We agree that both regions (Nino3 and Nino4) are however strongly interdependent, and that the effect of this mutual dependence is probably not sufficiently accounted for by our analysis. The additional consideration of the Nino3.4 region is just presented for the sake of

completeness, since it serves as the most commonly used reference region for defining indices for ENSO activity. All the aforementioned facts will be further explained in our revision. We would, however, prefer keeping the results for all three aforementioned regions explicitly in the manuscript for the reason described above unless removing parts of them (or shifting them to some Appendix) is deemed necessary by the handling academic editor.

*11. section 4.2: These results are interesting, but for their interpretation it would help to look at which specific changes have been observed (changes in atmospheric circulation, convection, etc.) during these periods.*

This is clearly a valid point, but would require either sophisticated speculations or very detailed additional analyses. We are strongly motivated to expand our corresponding manuscript by some more general qualitative discussions, but would prefer very specific analyses along the suggested lines of research to be addressed in some follow-up study to maintain the topical integrity of the present manuscript.

*12. p21, l25-27: The statement does not make sense (to me). Please rephrase.*

We will attempt to clarify this statement in the revised manuscript.

**Anonymous Referee #2**

*The authors investigate the connectivity properties of the temperature network with emphasis on the impacts of ENSO events and volcanic eruptions on the structure of the network.*

*While I have no problem with the methods and the analysis, I am not impressed with the results. That El Nino and La Nina have effects and long-range correlations on the temperature field is not news. The same goes with volcanic eruptions. I, therefore, agree with anonymous reviewer #1 that more has to be done for this paper to be an original paper.*

As Anonymous Referee #2 links his report to referee #1, we kindly refer to our comments on how we address the corresponding remarks by referee #1.

More specifically, the original contribution of this work lies in the application of a suite of network characteristics to study global patterns of climate "disruptions" that have not yet been used in combination for this purpose. In this context, we are confident that our work is

original. The point is not to show that El Nino and La Nina, as well as volcanic eruptions have large-scale impacts on global surface air temperatures (this would be trivial and is well established), but how these effects manifest in subtle structural properties of the spatio-temporal correlation structure of temperatures that are captured by our network measures. This goes beyond identifying teleconnection patterns, but rather addresses the way such teleconnections are generated (more regional versus global, smaller vs. larger scale spatially coherent patterns, etc.). To our best knowledge, this is an aspect that has not been previously covered in the literature and presents the novelty of our work. We will attempt to clarify this important point in a revision of our manuscript.

**References:**

J.F. Donges, I. Petrova, A. Loew, N. Marwan and J. Kurths, How complex climate networks complement eigen techniques for the statistical analysis of climatological data, Climate Dynamics 45(9), 2407-2424 (2015)

A. Gozolchiani, K. Yamasaki, O. Gazit, O. and S. Havlin, Pattern of climate network blinking links follows El Niño events, EPL (Europhysics Letters), 83, 28005 (2008)

J. Ludescher, A. Gozolchiani, M.I. Bogachev, A. Bunde, S. Havlin and H.J. Schellnhuber, Improved El Niño forecasting by cooperativity detection, Proceedings of the National Academy of Sciences, 110, 11742–11745 (2013)

J. Ludescher, A. Gozolchiani, M.I. Bogachev, A. Bunde, S. Havlin and H.J. Schellnhuber, Very early warning of next El Niño, Proceedings of the National Academy of Sciences, 111, 2064–2066 (2014)

A.H. Monahan, J.C. Fyfe, M.H.P. Ambaum, D.B. Stephenson and G.R. North, Empirical Orthogonal Functions: The Medium is the Message, Journal of Climate, 22, 6501–6514 (2009)

A. Radebach, R.V. Donner, J. Runge, J.F. Donges and J. Kurths, Disentangling different types of El Niño episodes by evolving climate network analysis, Physical Review E 88, 052807 (2013)

M. Wiedermann, A. Radebach, J.F. Donges, J. Kurths, and R.V. Donner, A climate network-based index to discriminate different types of El Niño and La Niña, Geophysical Research Letters 43(13), 7176–7185 (2016)

M. Wiedermann, J.F. Donges, R.V. Donner, D. Handorf, and J. Kurths, Hierarchical structures in Northern Hemispheric extratropical winter ocean-atmosphere interactions, International Journal of Climatology 37, 3821-3836 (2017)

K. Yamasaki, A. Gozolchiani and S. Havlin, Climate Networks around the Globe are Significantly Affected by El Niño, Physical Review Letters, 100, 228501 (2008)

---

## Editor Comment (EC1) · JM Restrepo (Editor) · 8 Apr 2018

The reviews of the mss point to major issues with the results (especially) and the methods presented in the paper. I will echo Ref #1's review: "My major point is therefore that the authors should make a better effort to connect this (network) description to physical phenomena found in observations and models. This requires a substantial literature search and a rewrite of the sections 2, 4 and 5."

What this entails is quantitative outcomes, in addition to qualitative ones. These quantitative outcomes should have meaning in the climate science context.

If the above is not successfully addressed, the referees will not likely allow this paper to proceed any further.

In addition to the points raised by Ref #1, there are editorial changes that require attention and should be dealt with in the next round of review: The Abstract has serious issues. For example: " Recent work has provided ample evidence that global climate dynamics at time-scales between multiple weeks and several years can be severely affected by the episodic occurrence of both, internal (climatic) and external (non-climatic) perturbations"

By recent, how many hundreds of years do you mean?

" we aim to improve our understanding...

we present an approach to

quantify teleconnectivity"

The first part says you aim to improve understanding of the physics. The second one state that you are going to present an approach...this makes no logical sense.

" we apply this

framework to study the impacts of different phases"

Impacts? you need to be specific.

"Taken together, the resulting time-dependent patterns

of network connectivity allow a tracing of the spatial extents of the dominating effects of both types of climate disruptions"

Since ENSO and volcanic teleconnections are known and there is quantitative techniques to eke these out already, you

either need to demonstrate that your method is superior to other methods and/or you need to tell us something new about these teleconnections.
[Figure]

In the Conclusions Section:

One conclusion is that you confirm what is known about ENSO already. The other is

"In this regard, one possible mechanism could involve the modulation of monsoons by strong El Niño and/or La Niña periods,

which could be further modulated by volcanic eruptions (Maraun and Kurths, 2005). Confirming this hypothesis in the context

of climate network studies would, however, require much more elaborated approaches than those used in the present work, and is therefore outlined as a subject of future research."

This reads as a speculation, not a conclusion, so it does not belong in the Conclusions Section.

Juan M. Restrepo

Handling Editor,

NPG